# A snapshot of selected neglected tropical disease research using the World Health Organization International Clinical Trials Registry Platform database, 1999–2023

Rhys Peploe [1,2]*, Hannah Jauncey[1,2], Yurika Sakai[1,2], Ghassan Karam[3], Anna Laura Ross[3], Sarah C. Charnaud[3], Prabin Dahal[1,2], Anthony W. Solomon[3], Philippe J. Guerin[1,2], Caitlin Naylor[1,2]

1 Infectious Diseases Data Observatory, University of Oxford, Oxford, United Kingdom, 2 Centre for Tropical Medicine and Global Health, Nuffield Department of Medicine, University of Oxford, Oxford, United Kingdom, 3 World Health Organization, Geneva, Switzerland

* rhys.peploe@ndm.ox.ac.uk

## Abstract

### Background

Knowledge of clinical study methodology and location can inform researchers, clinicians, funders, and policymakers of the gaps which exist in a disease's evidence landscape, thereby conferring guidance for appropriate resource allocation. This study summarises registered studies for selected neglected tropical diseases (NTDs) and identifies evidence gaps.

### Methodology

The International Clinical Trials Registry Platform (ICTRP) was searched in September 2023 to extract 315 clinical study registrations submitted between January 1999 and September 2023 on Chagas disease, schistosomiasis, soil-transmitted helminthiases, and visceral leishmaniasis. Data were standardised before a descriptive analysis was performed on study location, inclusion age range, study design, phase, and population access to studies.

### Results

Registered studies were partially aligned to the geographical distribution of disease burden; countries in which the highest number of studies were registered often face great burden, though there were several endemic countries with few or no studies. The number of registered studies increased over time in the period covered by the review, with 51–62% of studies across the 4 diseases conducted in 2014–23 period, as opposed to 42%-49% in the preceding 15 years. Only 12–17% of studies were multi-country studies across the 4 NTDs. 14% of Chagas disease studies included

**Data availability statement:** The data and code scripts used in this paper are publicly available at the following GitHub page: https://github.com/Infectious-Diseases-Data-Observatory/NTD-ICTRP-Landscape-Review.

**Funding:** This work was supported by the Gates Foundation (INV-021904 to GK, ALR, SCC, and sub-awarded from this grant to RP, HJ, YS, PJG, CN) and the Wellcome Trust (222410/Z/21/Z to RP, HJ, YS, PD, PJG, CN). AWS was unfunded. The Gates Foundation and Wellcome Trust had no role in study design, data collection and analysis, decision to publish, or preparation of the manuscript. Gates Foundation: https://www.gatesfoundation.org/ Wellcome Trust: https://wellcome.org/.

**Competing interests:** I have read the journal's policy and the authors of this manuscript have the following competing interests: GK, ALR, SCC and AWS are staff members of the World Health Organization. The authors alone are responsible for the views expressed in this article and they do not necessarily represent the views, decisions or policies of the institutions with which they are affiliated.

children under the age of 16 years (much lower than other NTDs) and only 2 studies (2%) exclusively studied under 16s.

## Conclusions

These findings highlight areas of research for these diseases which have been neglected between 1999–2023, indicating need for further research to fill these gaps and aid progress towards the World Health Organization's roadmaps to elimination of NTDs by 2030.

### Author summary

Neglected tropical diseases (NTDs) are a group of conditions which have devastating impacts on communities, particularly those in low- and middle-income countries where access to healthcare and treatments is often limited. Knowing when, where and how clinical studies investigating these diseases are being conducted allows researchers, clinicians, policymakers and funders to be aware of research gaps and inform decision-making. Consequently, people affected by these diseases benefit from the targeted research. In this study, we assessed four NTDs: Chagas disease, schistosomiasis, soil-transmitted helminthiases and visceral leishmaniasis using study registration data collated in the World Health Organization's International Clinical Trial Registry Platform. We identified several countries which are impacted by NTDs but have not been the location of any clinical studies, representing a gap in the knowledge regarding optimal treatment options and management of these diseases. Additionally, few studies researching Chagas disease included children, despite infections frequently occurring in the early years of life. Further research and a greater number of studies conducted in multiple countries should be undertaken to strengthen evidence in treatments and surveillance of these NTDs.

### Introduction

Neglected tropical diseases (NTDs) are a collection of diseases which severely impact vulnerable communities socially and economically. They cause substantial mortality and morbidity, whilst perpetuating poverty and inequality [1]. Over a billion people globally are affected by NTDs, predominantly, but not exclusively, in the most disadvantaged communities of low- and middle-income countries [2,3]. While progress has been made, many of the targets defined in the World Health Organization (WHO)'s 2012–2020 NTD roadmap [4] were not reached and progress against the 2021–2030 road map [5] remains slow and uneven [6]. Timely generation and dissemination of evidence and its incorporation in subsequent syntheses remains important to the collective ambitions to achieve these targets.

Clinical trial registries play a fundamental role in advancing science by providing a platform for studies to be registered; they promote research transparency, minimise publication bias and help to reduce research waste by avoiding duplication [7]. Since the first publicly accessible trial registry (ClinicalTrials.gov) was launched in 2000, several national and international trial registries have provided further platforms for investigators to register their studies [8–10], while directives from the International Committee of Medical Journal Editors (ICMJE) and others have encouraged study registration [11,12]. WHO developed the International Clinical Trials Registry Platform (ICTRP) [13] in 2005 to improve findability, increase interoperability and reduce heterogeneity; ICTRP collates records from multiple source registries that adhere to certain common standards, providing a single point of access [14]. Source registries which provide data to ICTRP include: ClinicalTrials.gov, European Union Clinical Trials Register (EU-CTR), Pan African Clinical Trial Registry (PACTR) and International Traditional Medicine Clinical Trial Registry, amongst others. Centralisation of records allows for a comprehensive and efficient method to assess the landscape of research and development (R&D). This is particularly valuable in the context of NTDs, which suffer from a chronic lack of R&D investment; associated research gaps remain numerous but are often difficult to track [15]. Assessing the location, characteristics and frequency of clinical studies focusing on NTDs provides researchers, clinicians, funders, and policy makers insights into the landscape of scientific efforts to control, eliminate and eradicate NTDs [16]. In addition to identifying research gaps, monitoring ongoing R&D activities informs decision-making for the progressive optimisation of limited resources.

This analysis explores clinical NTD research using the records indexed in the ICTRP database, with a focus on equity of research. Four diseases have been selected for an in-depth snapshot analysis: Chagas disease, schistosomiasis, soil-transmitted helminthiases (STH), and visceral leishmaniasis (VL). These four diseases were chosen as IDDO remains actively engaged with the research community across these diseases, maintains a systematic scoping review of trials landscape, has supported the development of a global research agenda, and currently maintains a data platform of individual level data. The objective is to assess whether clinical research aligned with the reported burden of disease and to identify research gaps.

## Methodology

### Data extraction & curation

The search to identify relevant records was completed in two stages; the eligibility criteria were pre-specified and agreed through discussions between IDDO and WHO. In the first stage, the ICTRP database was searched on 15 September 2023 to identify records on NTD which were published on or after 1 January 1999. Exclusion of records occurred if the condition was not an NTD, a vector study or human infection model where outcome was not related to disease or burden and if the focus of the study was not an NTD outcome, burden or treatment of an NTD. In the second stage, the records were further screened for the 4 targeted NTDs. These search terms are listed in S1 Text. Combinations of spellings and synonyms were used to maximise identification of relevant records. Studies were not restricted by language. The screening was conducted by a single author.

Several variables were extracted from the ICTRP database including disease assessed, year of study registration and enrolment, country, study phase, target sample size, inclusion criteria, exclusion criteria, source registry and several study design parameters. The extracted information underwent a series of standardisation procedures that included, for example, reformatting dates to align with the International Organization for Standardization (ISO) 8601 format [17] and transforming country names to ISO 3166 Country Codes [18]. Implementation of controlled terminology, use of subcategorisation, and manual extraction of relevant information from free text entries further enhanced data harmonisation.

### Disease burden, economic status, and region classification

Burden of each disease was represented by the mean annual Disability-Adjusted Life Years (DALYs) attributed to the disease per one million population, summarised at country level, taken from Institute for Health Metrics and Evaluation

(IHME) estimates [19] on 18 June 2025. Estimates for 2021 were the most recent data for all four diseases at the time of extraction. IHME used the term 'intestinal nematode infections' as opposed to soil-transmitted helminthiases; these terms have been considered synonymous.

Population size data for each country were obtained from the World Bank [20] on 14 November 2024. We used data from 2023, the most recent available at the time of extraction. Income group classification for each country was taken from the World Bank [21], via the worlddatr R package [22], and was accessed on 23 September 2025. The Bolivarian Republic of Venezuela did not have an income group classification due to the unavailability of data, and Ethiopia was placed in temporary unclassified status for the financial year 2026. However, since the temporary status was implemented after our analysis began, Ethiopia was classed as a low-income country for this analysis. WHO regions were used to group countries [23], while intermediate regions (i.e., Western Africa) and subregions (i.e., Sub-Saharan Africa) were defined by UN Geoscheme [24].

## Analyses

A descriptive analysis of studies in the ICTRP database registered between 1999–2023 was undertaken. Location of studies was compared with the burden and income group of the countries in which the studies took place. We summarised the geographical distribution of studies and disease burden over the study period (1999–2023) considered in this review and also present a further breakdown in two ten-year tranches (2004–2013 and 2014–2023) to assess temporal evolution of the research landscape. The inclusion age ranges of studies were analysed by WHO region. The distribution of multi-country studies by disease endemicity was assessed. Study phase was summarised by income status of the countries, with this analysis restricted to interventional, single-country studies.

*Population access analysis:* Increased access to clinical studies is positive for several reasons. Study validity is improved as the sample population is likely to be more representative and diverse, whilst also allowing more people to be a part of research which can influence treatments and guidelines. To investigate this on a country level, population access to studies was defined as the total targeted (planned) number of participants per 10,000 population of the country over the period covered in the review (1999–2023). Only interventional studies conducted in a single country were included for this analysis, as multi-country studies did not always indicate the proportion of the target sample size attributed to each country, preventing their inclusion.

## Software and analysis code

All analyses presented in this review were conducted using R statistical software [25]. The underlying dataset and R code are publicly available on the Infectious Disease Data Observatory (IDDO) GitHub [26].

## Results

The first stage ICTRP search identified 2,270 records for screening. Of these, 462 (20%) were excluded because the disease of interest was not an NTD at all, many of the excluded studies were Leukaemia studies, for instance. Further reasons for exclusions were: 12 (1%) were vector studies where the outcome was not disease- or disease-burden-related, 6 (<1%) were human infection models where the outcome was not disease- or disease-burden-related, and an additional 34 (1%) studies did not focus on an NTD outcome, burden or treatment. Of the remaining 1,756 records, 315 (18%) included one or more selected NTDs: Chagas disease ($n = 88$), schistosomiasis ($n = 102$), STH ($n = 48$) and VL ($n = 84$) (Fig 1). Of the 315 included for analysis, 7 (2%) assessed two of the four targeted NTDs (2 Chagas disease and schistosomiasis, 5 STH and schistosomiasis); these studies were included in the counts for each of their respective diseases (Table 1).

For each disease, between 51 and 62% of studies were conducted between 2014–2023, with the remaining studies registered in the preceding 15 years (1999–2013). The majority of studies (range across the 4 diseases: 70–83%) were interventional. This was as expected, since ICTRP is intended for clinical interventional studies for which registration is a

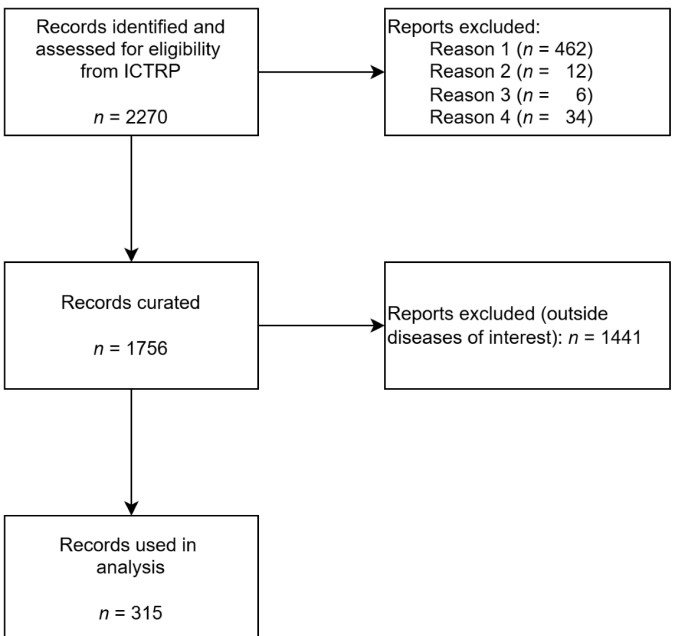

Registration screening flow.

Exclusion criteria:
 Reason 1: The condition studied is not an NTD
 Reason 2: Vector studies where outcome is not disease or burden related
 Reason 3: Human infection models where outcomes are not related to NTD disease or burden
 Reason 4: Focus of the study is not an NTD outcome, burden or treatment of NTD

**Fig 1. PRISMA flowchart of ICTRP record extraction.**

requirement [27]; observational studies made up between 17–30% of the studies and are generally not bound to the same registration requirements. One preventive schistosomiasis study and 3 bioavailability/bioequivalence (BA/BE) VL studies were identified. Interventional study design is presented by disease in S1 Table, which shows most registered interventional studies were randomised, parallel, and open-label studies.

Most studies were conducted in single countries. Less than one-fifth of studies aimed to enrol participants in multiple countries (Fig 2): 15% for Chagas disease (13/88), 12% for schistosomiasis (12/102), 17% for STH (8/48) and 14% for VL (12/84). There was no indication of increased numbers of multi-country studies in later phase trials (S2 Table). Multi-country studies in schistosomiasis tended to include high-income country (8 out of 12 studies), whereas VL multi-country studies were largely conducted in low- and lower-middle-income countries (11/12, 92%), and there was no discernible trend for STH and Chagas disease.

Of the interventional, single-country studies, study phase reporting varied across NTDs, with phase information clearly defined in 65% (34/52) of Chagas disease studies, 46% (33/71) of schistosomiasis studies, 26% (9/34) of STH studies and 77% (40/52) of VL studies (S1 Fig). Between 19–50% of the studies reported phases as 'not applicable', and a further 4–24% were missing, varying by disease. Early phase interventional studies (phase I) tended to be undertaken in

**Table 1. Study details. Note that column percentages may add to 99.9% or 100.1% due to rounding to one decimal point.**

|  | Chagas Disease | Schistosomiasis | Soil-transmitted helminthiasis | Visceral leishmaniasis |
|---|---|---|---|---|
| **Number of unique studies** | 88 | 102 | 48 | 84 |
| Total target participant size (studies reporting target size) | 69,919 (70) | 836,320 (99) | 165,869 (48) | 71,768 (81) |
| **Time period** |  |  |  |  |
| 1999-2003 | 2 (2.3%) | 0 | 0 | 1 (1.2%) |
| 2004-2013 | 37 (42.0%) | 39 (38.2%) | 20 (41.7%) | 40 (47.6%) |
| 2014-2023 | 49 (55.7%) | 63 (61.8%) | 28 (58.3%) | 43 (51.2%) |
| **Single vs multiple country studies** |  |  |  |  |
| Single country | 75 (85.2%) | 90 (88.2%) | 40 (83.3%) | 71 (84.5%) |
| Multiple country | 13 (14.8%) | 12 (11.8%) | 8 (16.7%) | 12 (14.3%) |
| Unclear | 0 | 0 | 0 | 1 (1.2%) |
| **Study type** |  |  |  |  |
| Interventional | 62 (70.5%) | 79 (77.5%) | 40 (83.3%) | 62 (73.8%) |
| Observational | 26 (29.5%) | 22 (21.6%) | 8 (16.7%) | 19 (22.6%) |
| Preventative | 0 | 1 (1.0%) | 0 | 0 |
| Bioavailability/Bioequivalence (BA/BE) | 0 | 0 | 0 | 3 (3.6%) |

high-income countries for schistosomiasis (5/8 studies) and VL (3/6 studies), while for Chagas disease and STH no phase I trials were performed in high-income countries. However, latter phase (phase II–IV) trials were conducted largely, in endemic countries, mostly in upper-middle- and lower-middle-income countries.

The median population access to clinical studies was lowest amongst high-income countries for all NTDs assessed (range: 0.010–0.15 per 10,000 population), followed by upper-middle-income countries (range: 0.01–0.78 per 10,000 population) (Fig 3). For Chagas disease, schistosomiasis and VL, lower-middle-income countries had the highest median target sample size per 10,000 amongst the income classifications, 1.0, 1.5, 0.12 respectively, while STH had a value of 0.89. The medians for low-income countries were 1.42 for schistosomiasis, 0.022 for VL and 56 for STH; the high value seen for STH is due to a study with a target sample size of 24,000 in Guinea-Bissau (112 target sample size per 10,000 population). High access to studies was also seen in Fiji (94 per 10,000, STH), Solomon Islands (81 per 10,000, STH) and Niger (53 per 10,000, schistosomiasis) (Fig 3).

## Chagas disease

The 88 Chagas disease studies had a total target enrolment size of 69,919 participants (median: 94, interquartile range (IQR): 45–354). Brazil had the highest total number of studies ($n = 42$, 48%), followed by Argentina ($n = 27$, 31%), Spain ($n = 14$, 16%), Bolivia and Colombia (both $n = 10$, 11%). Three of these countries were also in the 4 most burdened countries of 1999–2023: Bolivia, Brazil and Argentina (range: 425–1220 mean annual DALYs per million), whereas the

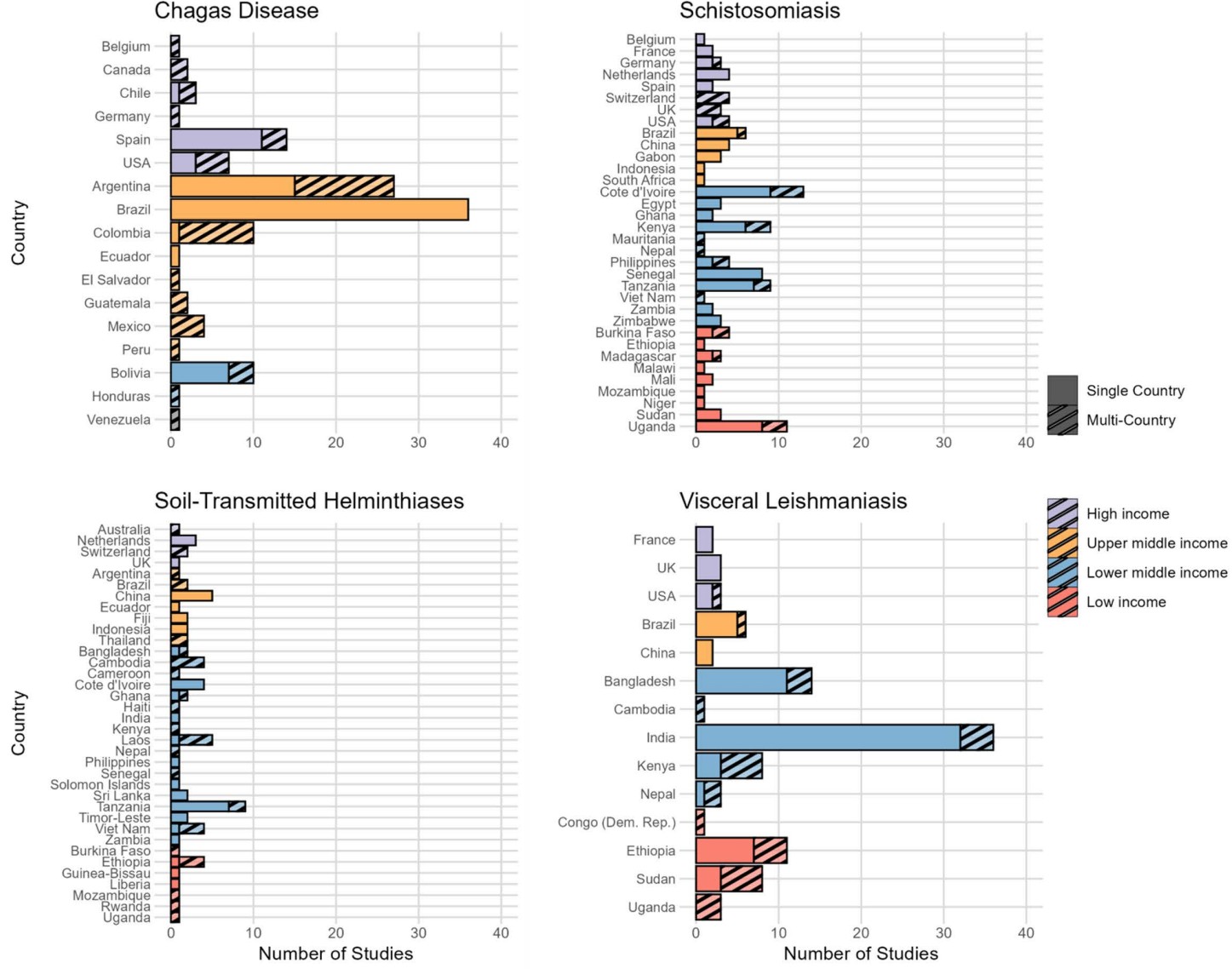

**Fig 2. Number of studies registered in ICTRP in 1999–2023, by country and disease.** Studies which occurred in multiple countries are represented by the striped bars and are counted individually by country. The income group of the host country is classified by the colour of the bars.

Bolivarian Republic of Venezuela had the third largest burden, 722 mean annual DALYs per million, though only one study registered (Fig 4). An average of 6 annual DALYs per million were recorded for Spain despite the relatively high number of single-country studies conducted. Of the 14 studies including Spain, 11 were single-country studies, with a median sample size of 280 (IQR: 64–900).

12 of the 88 studies (14%) had a minimum inclusion age under 16 years, contributed by 11 studies from the Region of the Americas and 1 from Europe (Fig 5). Of these, 2 also had a maximum inclusion age under 16 years, thereby assessing children specifically. 71 studies (81%) exclusively studied populations aged 16 years and over. Five studies (6%) had an unspecified inclusion age range.

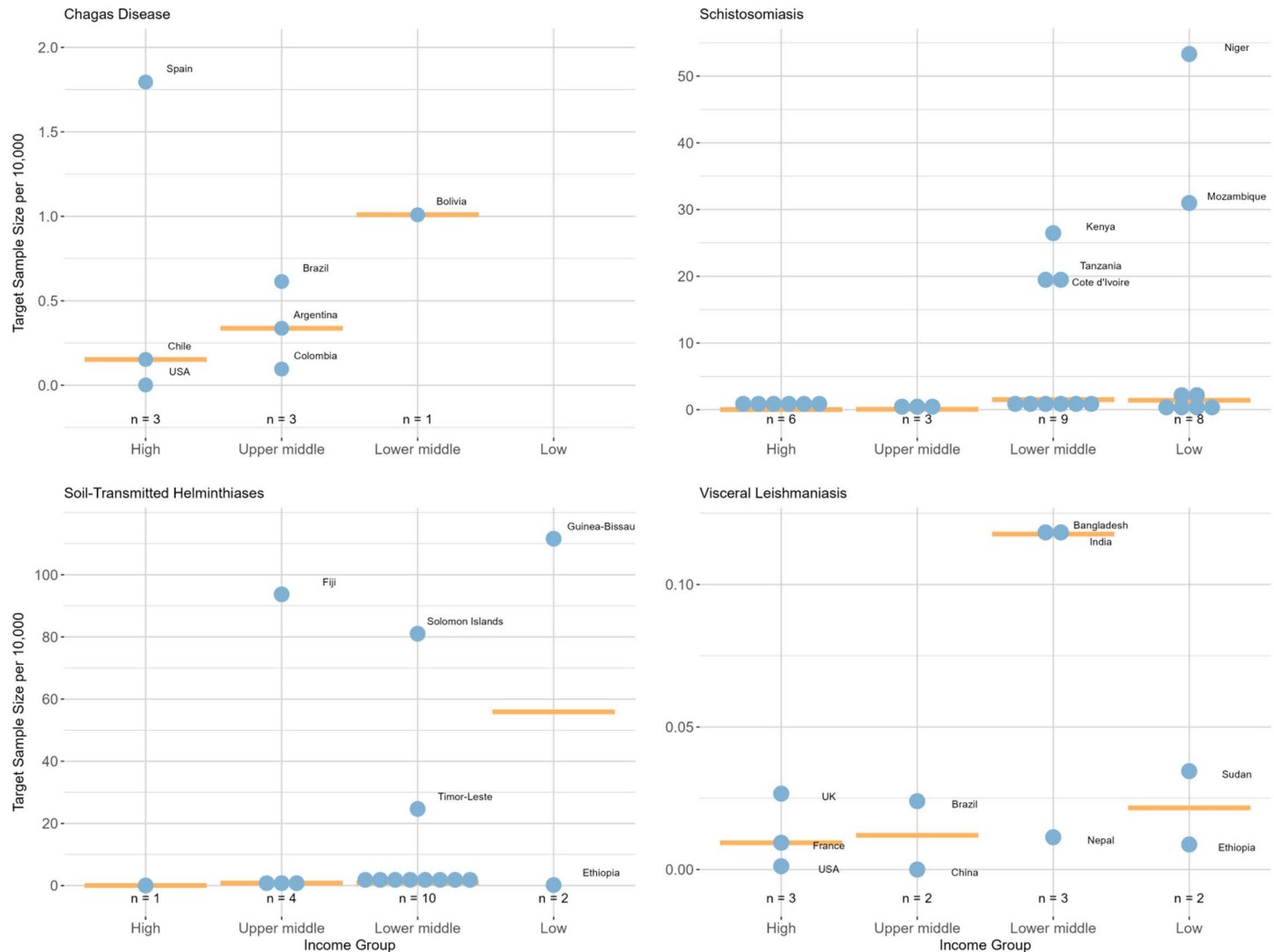

**Fig 3. Total target participant size per 10,000 population in countries where studies were registered in ICTRP in 1999–2023.** Median represented by the orange horizontal line. Only single-country interventional studies are included.

Of the 20 countries with the highest mean annual DALYs per million in 2004–2013, 12 had a decline in the mean annual DALYs per million over the next period, 2014–2023 (mean percent change 5%, range: –23% to 19%). Of the 12 countries that saw a decline in burden, 4 also saw a decrease in studies, three saw an increase in studies and 5 had no studies conducted in the country in either time period. Concurrently, the number of countries with studies conducted in 2004–2013 was 17, this fell to 8 in the following decade (2014–2023) (Fig 4). There was one study in Central America (Mexico) in 2014–2023, in contrast to 4 studies across 4 countries in 2004–2013. Burden in Central America increased in half (4/8) of the countries from 2004–2013 to 2014–2023, with a mean change of 1% (range: -5% to 19%). In Spain, the number of studies decreased from 9 to 5, while mean annual DALYs increased by 2% in 2014–2023 compared with 2004–2013. Conversely, more studies were recorded in Brazil, Argentina and Bolivia while disease burden reduced between

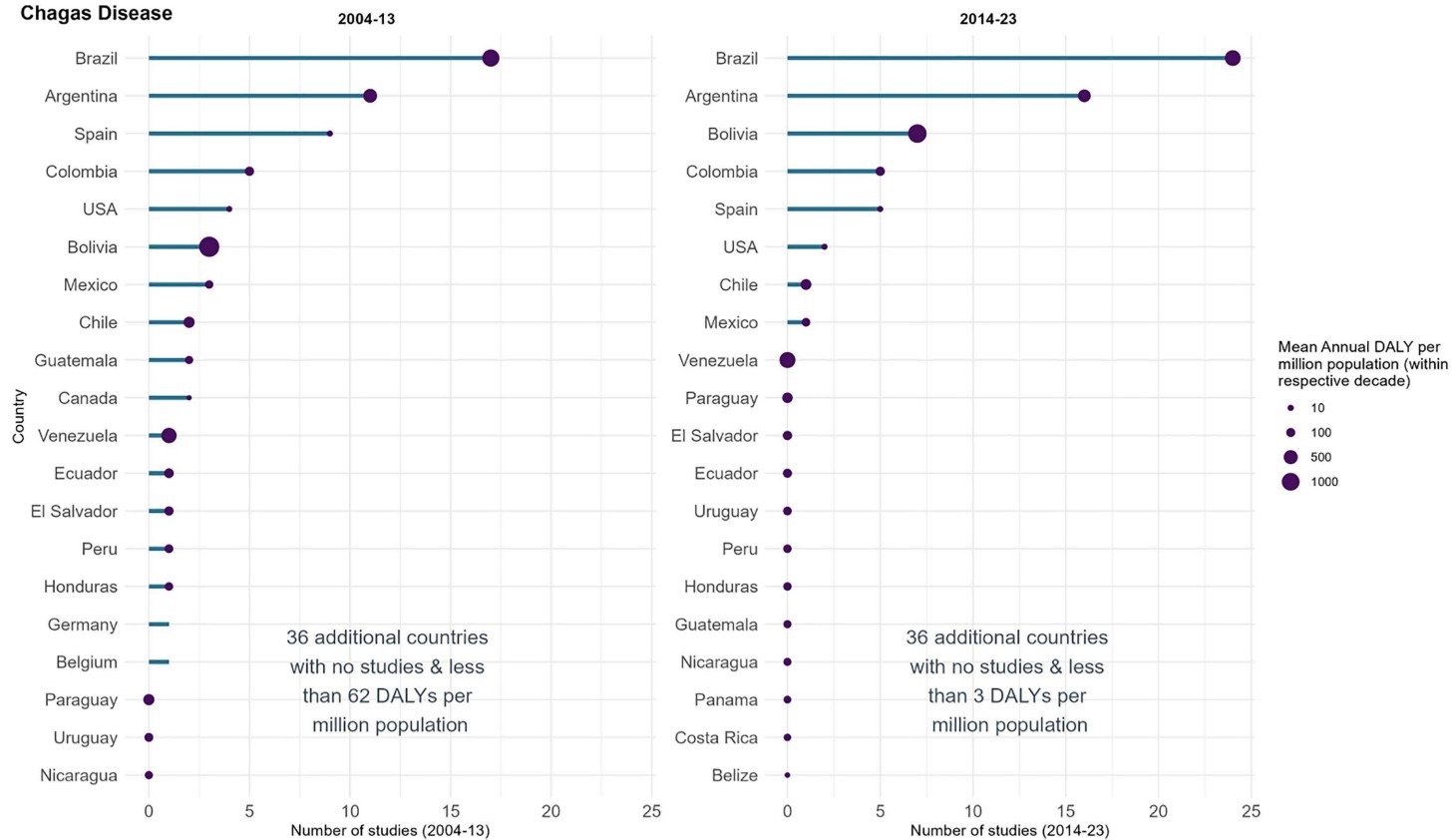

**Fig 4. Registered Chagas disease studies in ICTRP and disease burden by country, split by decade.** The length of the bar represents the number of studies registered in that country and decade in ICTRP. The size of the bubbles is proportional to the disease burden, represented as the mean annual DALYs per million population. The graph's y axis has been truncated after 20 countries for clarity.

2004–13 and 2014–23 (4–7 more studies, 21–23% burden decrease). Brazil or Argentina featured in 78% (38/49) of all studies in 2014–2023, increasing from 68% (25/37) in the previous decade.

## Schistosomiasis

102 schistosomiasis studies were identified, with a combined target participant size of 836,320 participants (median: 360, IQR: 103–1,285). The five countries with the greatest total number of studies conducted were Côte d'Ivoire (*n* = 13, 13%), Uganda (*n* = 11, 11%), Kenya (*n* = 9, 9%), United Republic of Tanzania (*n* = 9, 9%) and Senegal (*n* = 8, 8%) (Fig 6). These five countries had a mean annual DALY burden ranging from 1307 to 2225 per million population. Mauritius, the Central African Republic, Benin and Liberia had greater burden than the five countries listed above yet there were no studies from these countries in 1999–2023.

Two-thirds of the 102 studies (68, 67%) had a minimum inclusion age under 16 years (Fig 5). While 24 (24%) studies exclusively assessed those aged above 16 years, 10 studies had an unclear age range. All WHO regions had paediatric and adult populations included in studies.

The 30 countries with the highest mean annual DALY burden from 2004–2013 all had a fall in disease burden in 2014–2023 (mean percent change: -35%, range: –2% to –76%). Of these 30 countries, 2 also saw a decrease in the number of studies, 8 countries saw an increase, 3 the same number of studies and 17 did not have any studies in either

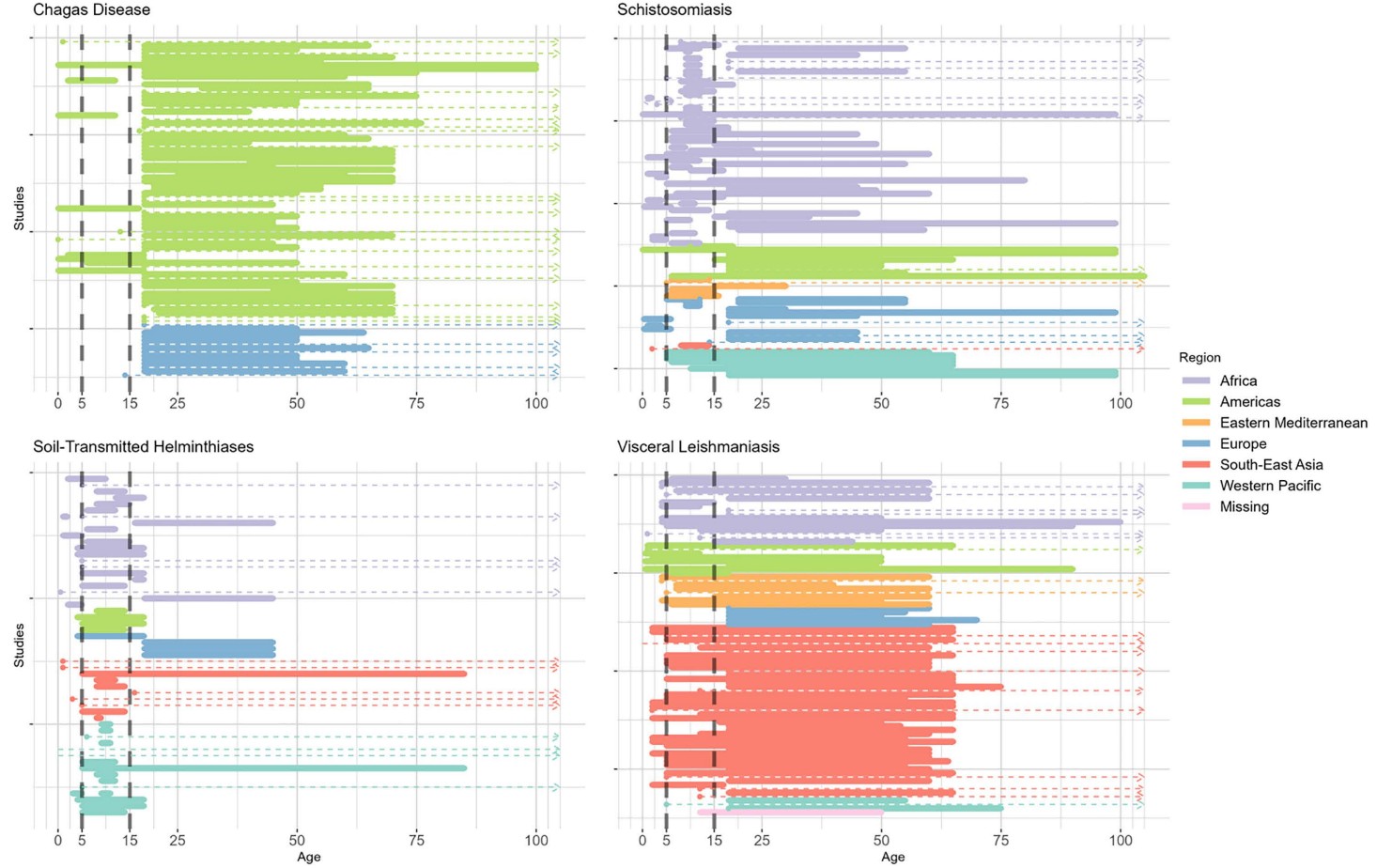

**Fig 5. Inclusion age ranges for studies registered in ICTRP in 1999–2023, by disease.** Colour indicates the WHO region of the study; studies across multiple regions are represented individually for each region. Horizontal dashed lines are used when one of the minimum or maximum values are not known, or there was no limit for inclusion. Vertical dashed lines are shown at 5 and 15 years.

time period. Studies were conducted in slightly greater number of countries in 2014–2023 (28 countries) than in 2004–2013 (25) (Fig 6). Four studies were registered in China in 2004–2013 (the equal highest number for a single country in that decade), however none were undertaken in 2014–2023; the disease burden in the two decades was 76 and 63 DALYs per million population. Four studies were registered in Côte d'Ivoire in 2004–2013, followed by 9 studies in 2014–2023, while burden decreased 45%, with mean annual DALYs falling from 2414 to 1327 per million. Senegal (2 to 6 studies, 51% decrease in burden) and the Netherlands (0 to 4 studies, no burden) also recorded an increase of studies between the two decades.

## Soil-transmitted helminthiases (STH)

48 studies assessed STH with a combined target participant size of 165,869 (median 1,194, IQR: 390–3,225). The United Republic of Tanzania (*n* = 9, 19%), China (*n* = 5, 10%) and Lao People's Democratic Republic (PDR) (*n* = 5, 10%) had the most studies during 1999–2023, with a mean annual DALY burden between 64 and 1,518 per million population (Fig 7). Conversely, 8 countries had a greater mean annual DALY per million (range: 1551–3423) than the United Republic of

PLOS Neglected Tropical Diseases

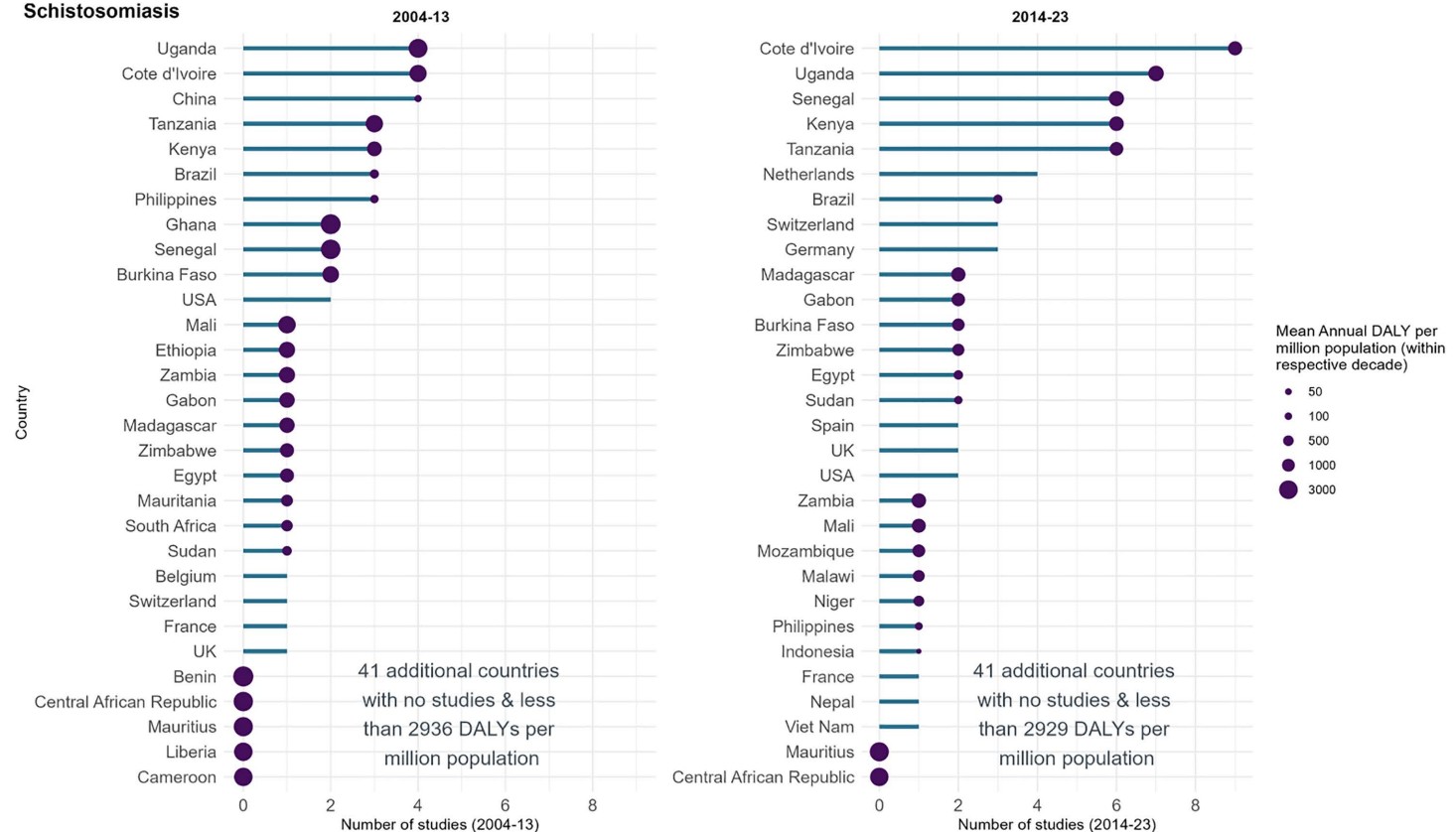

**Fig 6. Registered schistosomiasis studies in ICTRP and disease burden by country, split by decade.** The length of the bar represents the number of studies registered in that country and decade in ICTRP. The size of the bubbles is proportional to the disease burden, represented as the mean annual DALYs per million population. The graph's y axis has been truncated after 30 countries for clarity.

Tanzania yet were collectively involved in only two studies. The Netherlands ($n = 3$), Switzerland ($n = 2$), UK ($n = 1$) and Australia ($n = 1$) were involved in studies but considered to be STH-burden-free.

STH studies primarily assessed children (Fig 5), with 36 (75%) studies featuring minimum inclusion ages under 16 years. Five (10%) studies had unclear age ranges and the remaining 7 (15%) assessed ≥16-year-olds. Of the 36 studies assessing those aged under 16 years, half (18) also had a maximum age range under 16 years. Across the four studies in the Region of the Americas, the maximum inclusion age was 18 years, whereas 3 of the 4 European Region studies were exclusively in adult participants (aged ≥18 years).

The 30 countries with the highest mean annual DALY burden in 2004–2013 all had a lower mean annual DALY burden in 2014–23 (mean percent change: –43%, range: –78% to –7%). Of the 30 countries, studies declined in 2014–23 compared to 2004–13 in 3 countries, rose in 9 countries, saw no change in one country and no studies in 2004–23 in 17 countries. A greater number of countries were included in studies from 2014–2023 period (26 countries) than in 2004–2013 (18 countries) (Fig 7). China was the setting for 5 studies in 2004–13 during which the country had a mean annual DALY burden of 43 per million; the burden fell to 19 (-56%) in 2014–23 during which there were no clinical studies from the country. Lao PDR on the other hand had a mean annual DALY of 1584 per million during 2004–13 and no studies, whilst in 2014–23, there were 5 studies and the burden fell by 58% to an annual average of 661 DALYs. Moreover, 10 of the 18 studied countries in 2004–2013 did not conduct any studies in the following decade. As previously mentioned, three studies were

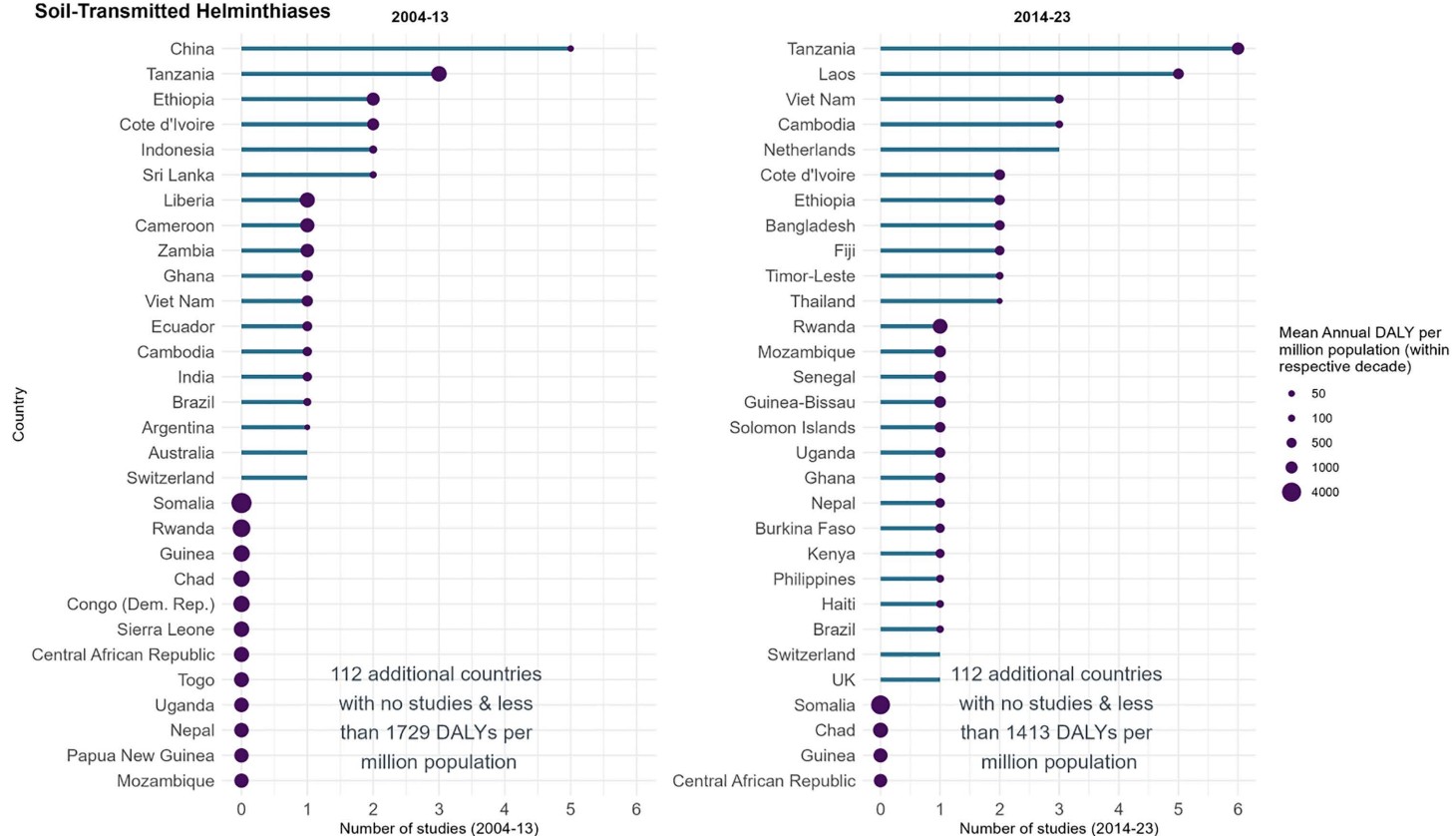

**Fig 7. Registered soil-transmitted helminthiases studies in ICTRP and disease burden by country, split by decade.** The length of the bar represents the number of studies registered in that country and decade in ICTRP. The size of the bubbles is proportional to the disease burden, represented as the mean annual DALYs per million population. The graph's y axis has been truncated after 30 countries for clarity.

completed in the Netherlands, all of which were conducted in 2014–2023 and were exclusively single-country studies (Fig 2).

### Visceral leishmaniasis (VL)

84 VL studies were identified, and the total participant size was 71,768 (median: 150, IQR: 64–566). The studies were predominantly from India ($n = 36$ studies, 43%), Bangladesh ($n = 14$, 14%), Ethiopia ($n = 11$, 11%), Kenya ($n = 8$, 10%) and Sudan ($n = 8$, 10%); these countries had disease burdens ranging from 133 to 950 mean annual DALYs per million population (Fig 8). Of these five countries, only Sudan was also one of the five highest-burdened countries, while the other four highest-burdened countries did not host any VL studies. The highest mean annual burden was South Sudan with 11,501 DALYs per million and no studies conducted. The UK ($n = 3$), USA ($n = 2$) and Cambodia ($n = 1$) conducted studies whilst having no identified VL burden. Country (or countries) could not be determined for one study, though this study was withdrawn.

Inclusion age ranges of studies often encompassed both children and adult populations, with 56 studies (67%) having a minimum inclusion age under 16 years, though only 4 of those 56 (7%) had a maximum inclusion age under 18 years (Fig 5). 23 (27%) of the 84 studies had a minimal inclusion age of 16 years or above, and 5 (6%) had unclear inclusion age information. All 5 European Region studies exclusively included adults (≥18 years).

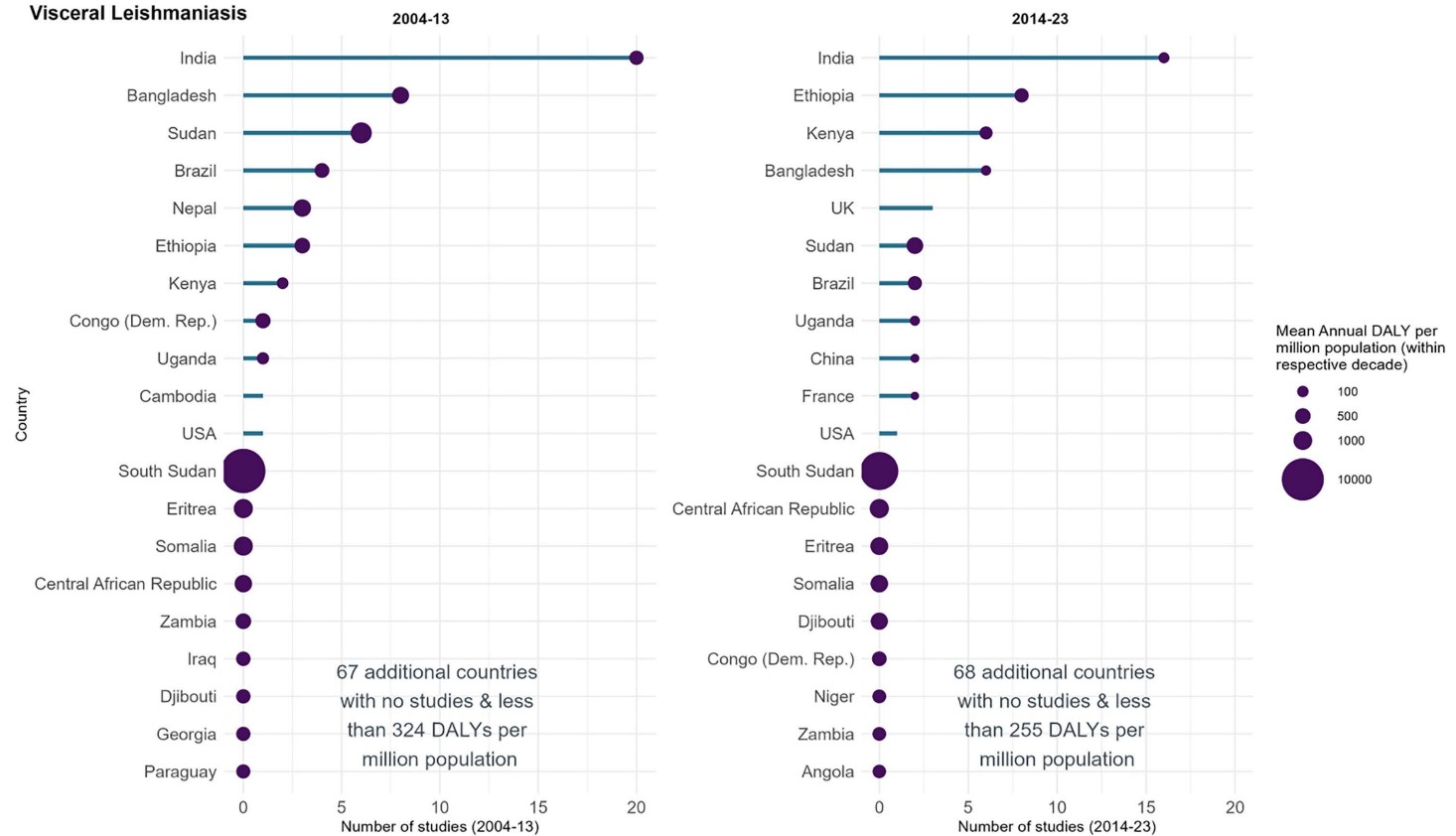

**Fig 8. Registered visceral leishmaniasis studies in ICTRP and disease burden by country, split by decade.** The length of the bar represents the number of studies registered in that country and decade in ICTRP. The size of the bubbles is proportional to the disease burden, represented as the mean annual DALYs per million population. The graph's y axis has been truncated after 20 countries for clarity.

17 of the 20 countries with the highest mean annual DALY burden in 2004–2013 saw a decrease in disease burden in 2014–2023 (mean percentage change -31%, range: –94% to 109%). Of the 17 countries, 6 also saw the number of studies decrease, whilst 2 saw an increase and 9 countries saw no studies in neither 2004–13 or 2014–23. In Southern Asia, 8 countries had some disease burden; all 8 saw a substantive decrease in mean annual DALYs per million with mean percent change of –66% (range: -16% to -94), while the number of studies decreased by 33%, from 33 in 2004–2013 to 22 in 2014–2023. The number of countries participating in studies in 2004–2013 remained the same in 2014–2023 (11 countries) (Fig 8). Ethiopia and Kenya had more studies in the latter decade (2014–23) than in 2004–2013, increasing from 3 and 2 to 8 and 6 studies, respectively. Ethiopia noted a 35% decrease in the mean annual DALYs per million from 495 (2004–2013) to 322 (2014–2023), Kenya saw an increase of 79% from 120 to 215. Sudan had a decrease in studies in 2014 compared to 2004–13, from 6 to 2 and a 55% decrease in mean annual DALYs (1461 DALYs per million to 653).

## Discussion

Our analyses demonstrate that for this selected group of NTDs, clinical studies listed within ICTRP were largely conducted in countries of significant disease burden. However, there were some high burden countries across all four diseases without any trials undertaken, likely indicating that many at-risk populations were not involved in clinical research. Conducting clinical research can be challenging in politically and economically unstable countries, which may explain the scarcity of

PLOS Neglected Tropical Diseases | https://doi.org/10.1371/journal.pntd.0014338   June 3, 2026

studies in some environments. There were also several countries, predominately high-income countries, which did not have a reported disease burden but were involved, partially or exclusively, in registered studies.

More study registration in all four NTDs in 2014–23 than in 2004–13 indicated a potential increase in the volume of research being implemented over time; it could also simply reflect an increase in the number of study registrations or due to the increased recognition of the importance of NTDs by the London Declaration of 2012 [28] and the WHO NTDs roadmaps [4,5], amongst other things. This has also coincided with a general global decline in the disease burden for each of the four NTDs considered (Figs 4 and 6–8). For example, there has been a vast reduction in VL disease burden in Southern Asia over the past 10 years [29] and a commensurate decline in the number of studies in this region, although India and Bangladesh remain amongst the most-researched VL populations. This reflects the region's programmatic commitment to elimination and scientific engagement [30], as well as significant investment from the public and private sector, including from the Gates Foundation and the Drugs for Neglected Diseases initiative (DNDi) [31]. The highest VL burdens worldwide are, at time of writing, in Sub-Saharan Africa, and there is an indication of an increase in studies being conducted in that region within the last decade, though this remains dwarfed by the VL R&D activity in Southern Asia. The recent launch of strategic frameworks and multi-country collaborations for the elimination of VL in Eastern Africa [32,33] could prompt continued increases in investment and research in the region, though this might be impacted by recent reductions in global official development assistance [34].

In the studies included in this review, there was also a predominance of single-country studies. A greater number of multi-country collaborations could enhance exchange of skills and resources, improve capacity, and enables undertaking high quality trial with robust external validity of the results. Owing to the larger generalisability of the results, multi-country studies would also avoid the need for generation of country-specific evidence required to guide national treatment policy, thus expediting access to drugs and diagnostics to patients who are most in need.

Our analysis also explored the distribution of the eligible age ranges of the participants across the 4 diseases. In general, all age-groups including children and adolescents were well represented in VL, STH and schistosomiasis, which largely matches the disproportionate burden of these diseases on young people [35–37]. However, for Chagas disease there was a conspicuous lack of paediatric studies (Fig 5). In Chagas disease, infections usually occur in the first years of life, though many cases are asymptomatic [38]. Patients suffering from chronic Chagas disease can then develop severe cardiac, digestive or neurologic conditions after 10–30 years [39,40]. Despite this, a conspicuous lack of paediatric Chagas disease studies was noted, suggesting limited research on the acute phase of the disease. This finding reflects the suggestion from a previous systematic review that <6% of Chagas disease studies solely recruited patients in the acute phase [41], though prevalent cases in children and younger adults has reduced since 1990 [42]. In contrast to the enrolment of patients in endemic countries, a quarter (10/40) of the studies in Europe were in healthy participants, and most of the studies included only adult populations.

Our analysis also investigated the population access to clinical studies across the 4 NTDs. While we found a greater access for people living in some endemic low- and lower-middle-income countries, recruitment only partially reflected underlying disease endemicity. In particular, the population access metric was driven by a handful of countries and studies, and most endemic countries had low population access. However, Spain, a high-income country, was an exception for Chagas disease where disease vectors are not native, but (due to migration) the disease burden is the greatest outside the Region of the Americas [43]. Low-income countries had relatively low population access to VL studies. This is important because these low-income countries, particularly Ethiopia and Sudan, are now the epicentre of the global VL burden. This weakness in the geographical distribution of clinical studies likely reflects major challenges in building health research capacity in LMICs [44] and highlights the need to enhance development of health research systems so that populations in need can access novel treatments earlier.

There are limitations to our review. First, we focused on only four NTDs, hence our results cannot necessarily be extrapolated to other NTDs. WHO is, at time of writing, leading work to develop a comprehensive R&D blueprint for NTDs;

such a blueprint would benefit from future similar evaluations across other NTDs regarding the current research agenda and highlight priority areas for R&D investment [45,46]. Second, there is likely a reporting bias, particularly in records before the 2004 ICMJE statement on study registration [11], so there are likely to be interventional studies completed in our period of review that were not listed in the ICTRP database. Though, compliance with registration continues to be imperfect [47]. Studies in registries that do not inform ICTRP will also have been overlooked by our analysis, as well as some observational studies which do not require registration. Thirdly, inclusion age ranges and target participant sizes may not necessarily reflect the actual population recruited. Furthermore, availability of data extracted by ICTRP from source registries limits the conclusions that can be made on some variables, such as trial type response and primary purpose. Finally, disease burden evolves constantly, the burden in one country or region can change dramatically in just a few years (S2–S5 Figs) or the burden within a country can also be concentrated in a handful of sub-regions; all these nuances have not been fully considered in this review.

In conclusion, the establishment of WHO's ICTRP was a major advance in creating a consolidated inventory of past and current clinical study registrations. This snapshot highlighted that the NTD study landscape in 1999–2023 partially aligns with disease burdens, but gaps exist for specific regions, sub-populations and study designs. In particular, there was limited research in a significant number of endemic countries, minimal studies in paediatric populations for Chagas disease, and low access to studies in some affected countries, particularly for VL. These gaps can be addressed by targeted funding. Increased research on these NTDs in 2014–2023 is positive, though it remains important that studies are not concentrated in a handful of lower-burden countries. Additionally, continued investment and collaborations towards VL elimination have been shown to be successful in Southern Asia. WHO's strategy on research for health asserts that "active national health research systems speed up the achievement of health goals" [48]. Analysis of the enablers and impact of the R&D ecosystem in resource limited countries could show how research by endemic countries can strengthen global research and drive reductions in NTDs. In the absence of robust systems to support research in endemic countries currently under-represented in clinical studies, researchers and developers must ensure that the products developed will be made accessible by, and be acceptable to, populations in which they have not been tested.

## Supporting information

**S1 Checklist. Prisma Checklist.**
(DOCX)

**S1 Text. ICTRP search terms for all NTDs.**
(DOCX)

**S1 Table. Design of interventional studies.** Note that column percentages may add to 99.9% or 100.1% due to rounding to one decimal point.
(DOCX)

**S2 Table. Centre status of study by phase.**
(DOCX)

**S1 Fig. Study phase by income group and disease.** Length of bars represent the number of studies for each study phase, with annotated totals and proportion of total studies by disease.
(TIF)

**S2 Fig. Ranking of Chagas disease burden by country over time.** Disease burden is represented as the annual DALYs per million. Each dot is one year; countries are linked via edges/lines. Selected countries are highlighted

in colours to show significant changes in ranking over the period 1999–2021 (2021 was the most recent burden data).
(TIF)

**S3 Fig. Ranking of schistosomiasis disease burden by country over time.** Disease burden is represented as the annual DALYs per million. Each dot is one year; countries are linked via edges/lines. Selected countries are highlighted in colours to show significant changes in ranking over the period 1999–2021 (2021 was the most recent burden data).
(TIF)

**S4 Fig. Ranking of soil-transmitted helminthiases burden by country over time.** Disease burden is represented as the annual DALYs per million. Each dot is one year; countries are linked via edges/lines. Selected countries are highlighted in colours to show significant changes in ranking over the period 1999–2021 (2021 was the most recent burden data).
(TIF)

**S5 Fig. Ranking of visceral leishmaniasis burden by country over time.** Disease burden is represented as the annual DALYs per million. Each dot is one year; countries are linked via edges/lines. Selected countries are highlighted in colours to show significant changes in ranking over the period 1999–2021 (2021 was the most recent burden data).
(TIF)

## Acknowledgments

Special thanks go to IDDO staff who have provided valuable insights and support to this research, especially Dr Sean Hackett and Tony Henry Oduor for their work on assessing data heterogeneity in the ICTRP database.

## Author contributions

**Conceptualization:** Rhys Peploe, Anna Laura Ross, Sarah C Charnaud, Prabin Dahal, Anthony W Solomon, Caitlin Naylor.

**Data curation:** Hannah Jauncey, Yurika Sakai.

**Formal analysis:** Rhys Peploe, Prabin Dahal.

**Funding acquisition:** Philippe J Guérin, Caitlin Naylor.

**Investigation:** Rhys Peploe, Caitlin Naylor.

**Methodology:** Rhys Peploe, Prabin Dahal, Caitlin Naylor.

**Project administration:** Caitlin Naylor.

**Software:** Rhys Peploe, Ghassan Karam.

**Supervision:** Prabin Dahal, Caitlin Naylor.

**Validation:** Rhys Peploe, Prabin Dahal.

**Visualization:** Rhys Peploe.

**Writing – original draft:** Rhys Peploe, Hannah Jauncey, Anna Laura Ross, Sarah C Charnaud, Prabin Dahal, Anthony W Solomon, Philippe J Guérin, Caitlin Naylor.

**Writing – review & editing:** Rhys Peploe, Hannah Jauncey, Yurika Sakai, Ghassan Karam, Anna Laura Ross, Sarah C Charnaud, Prabin Dahal, Anthony W Solomon, Philippe J Guérin, Caitlin Naylor.

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
