## [Decision Letter · Decision Letter 0]

26 Mar 2026

PNTD-D-26-00289

A snapshot of selected neglected tropical disease research using the World Health Organization International Clinical Trials Registry Platform database, 1999–2023

Dear Dr. Peploe,

Thank you for submitting your manuscript to PLOS Neglected Tropical Diseases. After careful consideration, we feel that it has merit but does not fully meet PLOS Neglected Tropical Diseases's publication criteria as it currently stands. Therefore, we invite you to submit a revised version of the manuscript that addresses the points raised during the review process.

We look forward to receiving your revised manuscript.

Kind regards,

David J. Diemert, M.D.

Academic Editor

Susan Madison-Antenucci

Section Editor

Shaden Kamhawi

co-Editor-in-Chief

Paul Brindley

co-Editor-in-Chief

**Journal Requirements:**

1) Please upload all main figures as separate Figure files in .tif or .eps format. For more information about how to convert and format your figure files please see our guidelines:

2) We have noticed that you have uploaded Supporting Information files, but you have not included a list of legends. Please add a full list of legends for your Supporting Information files after the references list.

3) Please amend your detailed Financial Disclosure statement. This is published with the article. It must therefore be completed in full sentences and contain the exact wording you wish to be published.

**Reviewers' comments:**

Reviewer's Responses to Questions

**Key Review Criteria Required for Acceptance?**

**Methods**

-Are the objectives of the study clearly articulated with a clear testable hypothesis stated?

-Is the study design appropriate to address the stated objectives?

-Is the population clearly described and appropriate for the hypothesis being tested?

-Is the sample size sufficient to ensure adequate power to address the hypothesis being tested?

-Were correct statistical analysis used to support conclusions?

-Are there concerns about ethical or regulatory requirements being met?

Reviewer #1: (No Response)

**Results**

-Does the analysis presented match the analysis plan?

-Are the results clearly and completely presented?

-Are the figures (Tables, Images) of sufficient quality for clarity?

Reviewer #1: (No Response)

**Conclusions**

-Are the conclusions supported by the data presented?

-Are the limitations of analysis clearly described?

-Do the authors discuss how these data can be helpful to advance our understanding of the topic under study?

-Is public health relevance addressed?

Reviewer #1: (No Response)

**Editorial and Data Presentation Modifications?**

Reviewer #1: (No Response)

**Summary and General Comments**

Reviewer #1: The authors present a review of studies conducted on four NTDs using the WHO International Clinical Trials Registry Platform. Through a systematic search of the platform, they identified all studies registered between 1999-2023 and extracted data on study characteristics for a descriptive analysis. The authors found that overall registered research has increased over time, but that study locations only partially overlapped with disease burden. As WHO is preparing an R&D framework for NTDs, this work clarifies gaps in the current evidence landscape that can be targeted with future work. As such I think this manuscript is a worthy contribution to the literature and have only the minor comments listed below.

Introduction

- Line 68 – include a clause that lists at least a few of the source registries that are included

- Lines 78-79 – why were these 4 NTDs selected for this review? Could the authors include a justification?

Methods

- Line 83 – data extraction section – were eligibility criteria for included studies pre-specified?

- Line 83 – data extraction sectionDid the authors consider collecting data on whether or not results were reported? Or otherwise, study status in general? It could be useful to know what percentage of studies that should have reported results indeed have reported results. Or the authors could clarify that results data are not included in the WHO registry in the same way that they are in others, like clinicatrials.gov in more recent years

- Line 100 – disease burden, economic status, region classification section – for each indicator, please classify what geographic unit is being referred to. For example, based on the results, it looks like these indicators were extracted for countries specifically and later grouped into regions

- Line 124 – I didn’t follow what “population access” meant in this context – could the authors elaborate?

Results

- Lines 134-135 – did the authors mean that the 462 articles were not focused on an NTD of interest (meaning, one of the 4 selected for this study)? Or not focused on NTDs at all?

- Figure 4 – could the authors include more bubbles on the legend to give a sense of the size being represented by the smaller bubbles? The text indicates that Spain had only 6 annual DALYs per million, but the legend only gives sizes for 500 and 1000

- Lines 222-225 – did the 12 countries that saw declines in DALYs overlap with the decline in studies (did the same countries see fewer studies that saw declines in DALYs)?

Discussion

- Line 336 – now known as the Gates Foundation (no longer Bill & Melinda Gates Foundation)

- Lines 374-377 – I didn’t quite follow this logic. It seems to me that conducting a similar evaluation now for similar NTDs would support the development of the R&D framework by identifying gaps, as the authors have stated in the objectives previously, rather than the other way around

PLOS authors have the option to publish the peer review history of their article (what does this mean?). If published, this will include your full peer review and any attached files.

**Do you want your identity to be public for this peer review?** For information about this choice, including consent withdrawal, please see our Privacy Policy.

Reviewer #1: No

**Figure resubmission:**
---

## [Decision Letter · Decision Letter 1]

5 May 2026

Dear Mr Peploe,

We are pleased to inform you that your manuscript 'A snapshot of selected neglected tropical disease research using the World Health Organization International Clinical Trials Registry Platform database, 1999–2023' has been provisionally accepted for publication in PLOS Neglected Tropical Diseases.

Best regards,

David J. Diemert, M.D.

Academic Editor

Susan Madison-Antenucci

Section Editor

Shaden Kamhawi

co-Editor-in-Chief

Paul Brindley

co-Editor-in-Chief

Reviewer's Responses to Questions

**Key Review Criteria Required for Acceptance?**

**Methods**

-Are the objectives of the study clearly articulated with a clear testable hypothesis stated?

-Is the study design appropriate to address the stated objectives?

-Is the population clearly described and appropriate for the hypothesis being tested?

-Is the sample size sufficient to ensure adequate power to address the hypothesis being tested?

-Were correct statistical analysis used to support conclusions?

-Are there concerns about ethical or regulatory requirements being met?

Reviewer #1: (No Response)

**Results**

-Does the analysis presented match the analysis plan?

-Are the results clearly and completely presented?

-Are the figures (Tables, Images) of sufficient quality for clarity?

Reviewer #1: (No Response)

**Conclusions**

-Are the conclusions supported by the data presented?

-Are the limitations of analysis clearly described?

-Do the authors discuss how these data can be helpful to advance our understanding of the topic under study?

-Is public health relevance addressed?

Reviewer #1: (No Response)

**Editorial and Data Presentation Modifications?**

Reviewer #1: (No Response)

**Summary and General Comments**

Reviewer #1: The authors have fully responded to my original review and I have no further comments. Thanks for the opportunity to review this interesting project.

PLOS authors have the option to publish the peer review history of their article (what does this mean?). If published, this will include your full peer review and any attached files.

Reviewer #1: No

---

## [Editor Report · Acceptance letter]

Dear Mr Peploe,

We are delighted to inform you that your manuscript, "A snapshot of selected neglected tropical disease research using the World Health Organization International Clinical Trials Registry Platform database, 1999–2023," has been formally accepted for publication in PLOS Neglected Tropical Diseases.

Best regards,

Shaden Kamhawi

co-Editor-in-Chief

Paul Brindley

co-Editor-in-Chief
